# Risk factors in adolescents as predictors of arterial hypertension in adults: Protocol for a systematic review

Márcia Gisele Santos da Costa[1]*, J. Jaime Caro[2], Katia Vergetti Bloch[1]

**1** Instituto de Estudos de Saúde Coletiva (IESC), Universidade Federal do Rio de Janeiro, Rio de Janeiro, Rio de Janeiro, Brazil, **2** Faculty of Medicine, McGill University, Montreal, Quebec, Canada

☯ These authors contributed equally to this work.

* mgisele@gmail.com

## Abstract

This is a protocol of a review paper, and there is no abstract. This review is part of a doctoral project that aims to develop a discrete event simulation model to predict how many adolescents may become hypertensive in adulthood. We will use data from the Brazilian study of cardiovascular risks in adolescents, called ERICA (Portuguese acronym). This study may help promote adherence to disease prevention protocols.

**Funding:** The author(s) received no specific funding for this work.

**Competing interests:** The authors have declared that no competing interests exist.

## Introduction

Cardiovascular diseases have been the leading cause of death in Brazil for more than three decades. Early control of high blood pressure (HBP), smoking, hypercholesterolemia, and obesity may contribute to the reduction of cardiovascular morbidity and mortality [1].

HBP is defined by sustained blood pressure (BP) levels above systolic 140 or diastolic 90 mmHg (millimeters of mercury), or both for adults, according to the 7th Brazilian Guidelines for Hypertension [1]. It is a multifactorial clinical condition often associated with functional or structural alterations of the target organs (heart, brain, kidneys, and blood vessels); and metabolic changes, with a consequent increase in the risk of fatal and non-fatal cardiovascular events [1, 2]. HBP affects 30% of the adult population, and about one third do not know their condition. According to the World Health Organization (WHO) [2], 17 million people die each year worldwide due to cardiovascular diseases. HBP increases the risk of myocardial infarction, stroke, and renal failure, and can cause heart failure. It is an asymptomatic disease, and many individuals are diagnosed only when the first complication arises. Factors associated with the development of hypertension include age, sex, ethnicity, body weight, salt and alcohol intake, lifestyle, socioeconomic status, and genetic factors [3].

Interest in the assessment of BP in children and adolescents began in the 1960s. The first recommendations for routine BP measurement in this age group appeared only in the 1970s. Until then, only secondary changes in BP were identified. New studies have emerged to understand the behavior of BP in this age group, its determining factors, its relationship with future HBP or cardiovascular disease, and to adopt primary prevention measures [4].

Aspects such as eating habits, lifestyle, and high blood pressure in childhood and adolescence, are important in the occurrence of cardiovascular diseases in adults [5]. Some risk factors, such as overweight, physical inactivity, and inadequate diet, are now more prevalent than in previous times in children and adolescents [5].

A systematic review estimated the pooled prevalence of HBP in Latin America in 6.2% (95% CI 3.1–10.6) [6]. Another study estimated the pooled prevalence of HBP in Brazilian adolescents in 8.12% (95% CI 6.24–10.52) [7].

It is essential to highlight the difference between association and prediction. An association indicates whether the exposition is more likely to be present in individuals who have hypertension, but doesn't necessarily imply that it is a causal risk factor. More important, if it is not causal, then measures to control it will have no effect [8].

It has been shown that HBP in childhood can be a predictive factor for systemic arterial hypertension in adulthood. Several cohort studies have found a significant correlation between HBP in children and adolescents and hypertension in adulthood [9, 10]. Children with blood pressure above the 90th percentile are 2.4 times more likely to be hypertensive adults [11]. Although essential hypertension in children is not a risk factor for cardiovascular events in childhood, cardiovascular and hemodynamic changes can be seen in these individuals from the second decade of life or even earlier [12].

Question to be answered: What are the predictors in childhood of HBP in adulthood.

The objective of this review is to assess the published evidence on risk factors for HBP present in adolescents and their contribution to the development of the disease in adulthood.

## Materials and methods

This systematic review was registered in PROSPERO ID: CRD42020172254. This protocol for systematic review follows the principles recommended in PRISMA-P for protocols [13] (S1 Checklist).

### Eligibility criteria

The studies will be selected according to the following criteria:

**Types of studies.** *Cohort studies*. We will include prospective and retrospective cohort studies that measured risk factors in adolescence and document their long-term association with HBP occurring in adulthood.

**Participants.** Studies examining adolescents (aged 12 to 17 years), in general population.

**Outcome measure.** The outcome is the relative risk (RR) or other measure of association (OR, HR) of HBP in adulthood (age over 18 years). for each of the risk factors in adolescence. If the outcome measure is another (OR, for example), the results will be grouped according to the measure presented, or converted to RR.

### Exclusion criteria

Studies focused on specific populations known to have a higher incidence of HBP (such as diabetics, adolescents who are obese, or have cardiac, or renal diseases).

**Search methods for identification of studies.** The databases to be searched will include Embase, LILACS, ADOLEC, MEDLINE, Cochrane Library, and references of selected articles that meet the inclusion criteria. There will be no restriction on language and date. The terms (and respective entry terms) used will be: risk factor, adolescent, trends, hypertension, high blood pressure, adults, middle aged, epidemiologic studies, cohort studies, follow-up studies, longitudinal studies.

**Data collection and analysis.**   *Selection of studies.* Two reviewers will independently screen identified titles and abstracts applying the eligibility criteria described above. Full texts will then be analyzed to determine which studies meet the defined inclusion criteria. The selection will be made using Rayyan QCRI [14], which is a 100% free web/mobile application. Disagreements at this stage will be resolved by consensus or by involving a third reviewer.

*Data extraction and management.* The extraction will be carried out by two peers independently using a standard form. When more than one publication of a study is available, the most recent data will be used, unless relevant results were published in an earlier version. Disagreements will be resolved by consensus or by involving a third reviewer.

**Predictive factors.**   The potential predictive factors are defined by the guidelines for hypertension of the Brazilian Society of Cardiology [1]: sex, ethnicity, body weight (overweight and obesity), salt and alcohol intake, physical activity, family history and socioeconomic factors. If found in a paper reported other factors not listed with a strong association for adolescents, these will be included.

*Quality assessment.* The methodological quality of potential studies will be assessed using the Newcastle-Ottawa scale (NOS) for non-randomized studies. We will use the "Quality In Prognosis Studies" (QUIPS) [15] tool that consists of several prompting items categorized into six domains ((1) study participation, (2) study attrition, (3) prognostic factor measurement, (4) outcome measurement, (5) study confounding, and (6) statistical analysis and reporting), and each domain is judged on a three-grade scale (low, moderate or high risk of bias).

## Data analysis and statistical considerations

Data synthesis and analyses will be done using the Cochrane Review Manager software, RevMan 5.4. When possible (depending on the number of studies), studies will be combined in a meta-analysis. For variables, the association will be expressed as RR, or some other measure, for occurrence of HBP in adults. Uncertainty will be quantified with a 95% CI. The heterogeneity between the studies will be assessed using the $I^2$ statistic. When $I^2 < 50\%$, we will use the fixed-effect model. In the event of significant inconsistency, possible causes will be assessed on a case-by-case basis and, when necessary, subgroup analysis will be carried out. We will use funnel plots to assess the potential existence of reporting bias. If there is evidence of effect modification, the analysis will be stratified by sex, ethnicity, and socioeconomic factors.

## The status of the study

The study is in the data collection and analysis phase. The initial deadline for completion is October2021.

## Supporting information

**S1 Checklist. PRISMA-P-checklist RS risk factors.**
(DOCX)

## Author Contributions

**Conceptualization:** Márcia Gisele Santos da Costa, J. Jaime Caro, Katia Vergetti Bloch.

**Data curation:** Márcia Gisele Santos da Costa.

**Formal analysis:** Márcia Gisele Santos da Costa.

**Investigation:** Márcia Gisele Santos da Costa.

**Methodology:** Márcia Gisele Santos da Costa, J. Jaime Caro, Katia Vergetti Bloch.

**Project administration:** Márcia Gisele Santos da Costa.

**Supervision:** J. Jaime Caro, Katia Vergetti Bloch.

**Validation:** J. Jaime Caro, Katia Vergetti Bloch.

**Writing – original draft:** Márcia Gisele Santos da Costa.

**Writing – review & editing:** Márcia Gisele Santos da Costa, J. Jaime Caro, Katia Vergetti Bloch.

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
