## [Decision Letter · Decision Letter 0]

24 Jun 2021

PONE-D-21-04728

Risk factors in adolescents as predictors of arterial hypertension in adults: protocol for a systematic review

PLOS ONE

Dear Dr. Costa,

Thank you for submitting your manuscript to PLOS ONE. After careful consideration, we feel that it has merit but does not fully meet PLOS ONE’s publication criteria as it currently stands. Therefore, we invite you to submit a revised version of the manuscript that addresses the points raised during the review process.

The Study Protocol format aims to support a strong methodological approach, increase the reproducibility of results, and address publication bias. Authors first submit a Study Protocol that should include a study’s rationale, its timeline, and proposed methodology for data collection and analysis; this article type does not include results, although it may report pilot data. The authors may disregard reviewer 1’s comments regarding lack of results. They need to carefully address the other reviewer comments.

We look forward to receiving your revised manuscript.

Kind regards,

Johannes Vogel

Academic Editor

PLOS ONE

Journal Requirements:

Reviewers' comments:

Reviewer's Responses to Questions

**Comments to the Author**

1. Does the manuscript provide a valid rationale for the proposed study, with clearly identified and justified research questions?

Reviewer #1: Yes

Reviewer #2: Yes

2. Is the protocol technically sound and planned in a manner that will lead to a meaningful outcome and allow testing the stated hypotheses?

Reviewer #1: Partly

Reviewer #2: Yes

3. Is the methodology feasible and described in sufficient detail to allow the work to be replicable?

Reviewer #1: No

Reviewer #2: Yes

4. Have the authors described where all data underlying the findings will be made available when the study is complete?

Reviewer #1: No

Reviewer #2: Yes

5. Is the manuscript presented in an intelligible fashion and written in standard English?

Reviewer #1: Yes

Reviewer #2: Yes

6. Review Comments to the Author

You may also provide optional suggestions and comments to authors that they might find helpful in planning their study.

Reviewer #1: This manuscript contains a plan to review literature to identify childhood predictors of adult hypertension. However, no data was presented. I am unclear as to the rationale for submitting this preliminary work for publication given that there are no results and nothing meaningful will be contributed to the published literature on this subject.

Reviewer #2: Pg 1 – Please explain numbers 1 and 3 in the affiliation - for the first and last authors.

Pg 2- Background – P 2 – “HBP is defined by sustained blood pressure (BP) levels above systolic 140 or diastolic 90 mmHg (millimeters of mercury), or both.” – Please review the definition.

In accordance with most major guidelines it is recommended that hypertension be diagnosed when a person’s systolic blood pressure in the office is ≥ 140 mm Hg and/or their diastolic blood pressure is ≥ 90 mm Hg. This definition does not apply to the United States. In the 2017 ACC/AHA guideline, hypertension is defined as BP ≥130 and / or ≥80 mm Hg.

Another important thing is that these definitions apply to adults ≥ 18 years of age.

For the adolescents, what definition do you intend to use?

Pg 5 - If the included population consists of adolescents aged between 12 to 17 years and the outcomes are measured at adulthood (age over 19 years), how are considered subjects aged 18 years – adolescents or adults?

Pg 5 row 108 – a bracket is missing

Pg 5 row 113 – a space and a comma in addition

Pg 5 – Search methods – “The databases to be searched will include Embase, LILACS, ADOLEC, MEDLINE, Cochrane Library, and bibliographic citations. “ What will be the selection criteria for “bibliographic citations” ?

Pg 5 - There may be too many search terms.

I recommend not using the term “skin pigmentation” ; this is not a risk factor for the purpose of this review.

Pg 7 – “The initial deadline for completion is April 2021.”

A new deadline must be set.

7. PLOS authors have the option to publish the peer review history of their article (what does this mean?). If published, this will include your full peer review and any attached files.

Reviewer #1: No

Reviewer #2: No

---

## [Author Response · Author response to Decision Letter 0]

4 Aug 2021

Response to Reviewers

Response: done.

Response: reference 4 corrected.

Response: Please make the following change regarding the availability of data: The results of this study will be available in article format to be submitted for publication.

Response: done (page 4, line 92 and after references).

Reviewer #1: 

This manuscript contains a plan to review literature to identify childhood predictors of adult hypertension. However, no data was presented. I am unclear as to the rationale for submitting this preliminary work for publication given that there are no results and nothing meaningful will be contributed to the published literature on this subject.

Response written by the editor: Authors first submit a Study Protocol that should include a study’s rationale, its timeline, and proposed methodology for data collection and analysis; this article type does not include results, although it may report pilot data. The authors may disregard reviewer 1’s comments regarding lack of results. 

Reviewer #2:

Pg 1 – Please explain numbers 1 and 3 in the affiliation - for the first and last authors.

Response: We apologize for the typo. The first and last authors are from the same institution. Affiliation has been corrected (number 3 has been removed and inserted number 1 for last author)

Pg 2- Background – P 2 – “HBP is defined by sustained blood pressure (BP) levels above systolic 140 or diastolic 90 mmHg (millimeters of mercury), or both.” – Please review the definition.

In accordance with most major guidelines it is recommended that hypertension be diagnosed when a person’s systolic blood pressure in the office is ≥ 140 mm Hg and/or their diastolic blood pressure is ≥ 90 mm Hg. This definition does not apply to the United States. In the 2017 ACC/AHA guideline, hypertension is defined as BP ≥130 and / or ≥80 mm Hg.

Another important thing is that these definitions apply to adults ≥ 18 years of age.

For the adolescents, what definition do you intend to use?

Response: We use the definition of HBP according to the 7th Brazilian Guidelines for Hypertension for adults (included in the text). For the adolescents, in this review we will identify the definition used by the authors of each selected study.

Pg 5 - If the included population consists of adolescents aged between 12 to 17 years and the outcomes are measured at adulthood (age over 19 years), how are considered subjects aged 18 years – adolescents or adults?

Response: This has been corrected as the subjects aged 18 years will be considered adults (included in line 104 page 5). 

Pg 5 row 108 – a bracket is missing

Response: corrected

Pg 5 row 113 – a space and a comma in addition 

Response: corrected

Pg 5 – Search methods – “The databases to be searched will include Embase, LILACS, ADOLEC, MEDLINE, Cochrane Library, and bibliographic citations. “ What will be the selection criteria for “bibliographic citations” ?

Response: Bibliographic citations refers to the references provided in the articles that meet the inclusion criteria. Corrected in the text.

Pg 5 - There may be too many search terms.

I recommend not using the term “skin pigmentation”; this is not a risk factor for the purpose of this review. 

Response: We agree and removed this term from the search.

Pg 7 – “The initial deadline for completion is April 2021.” A new deadline must be set. 

Response: Amended to October

---

## [Editor Report · Decision Letter 1]

11 Aug 2021

Risk factors in adolescents as predictors of arterial hypertension in adults: protocol for a systematic review

PONE-D-21-04728R1

Dear Dr. Costa,

We’re pleased to inform you that your manuscript has been judged scientifically suitable for publication and will be formally accepted for publication once it meets all outstanding technical requirements.

Kind regards,

Johannes Vogel

Academic Editor

PLOS ONE
---

## [Editor Report · Acceptance letter]

12 Aug 2021

PONE-D-21-04728R1 

Risk factors in adolescents as predictors of arterial hypertension in adults: protocol for a systematic review 

Dear Dr. Costa:

I'm pleased to inform you that your manuscript has been deemed suitable for publication in PLOS ONE. Congratulations! Your manuscript is now with our production department. 

Kind regards, 

on behalf of

Professor Johannes Vogel 

Academic Editor

PLOS ONE